# Design and Simulation of a High-Speed Star Tracker for Direct Optical Feedback Control in ADCS

**DOI:** 10.3390/s20082388

**Published:** 2020-04-22

**Authors:** Mikaël Marin, Hyochoong Bang

**Affiliations:** Aerospace Systems and Control Laboratory, Korea Advanced Institute of Sciences and Technologies, Daejeon 34141, Korea; marin.mikael@ascl.kaist.ac.kr

**Keywords:** star tracker, star detection, high-speed, IMU-free, error correction, attitude determination, ADCS, CubeSat, computer vision, image processing, FPGA

## Abstract

Star Trackers are often the most accurate instrument in an Attitude Determination and Control Systems, but often present a slow update rate, requiring additional sensor and sensor fusion algorithms to provide a smoother and faster output. However, the available rate gyros are either noisy, or expensive and heavy. The proposed work investigates the feasibility of high-speed star trackers with modern optics, sensors, and computing systems. Firstly, we investigate the sensitivity of an optoelectrical acquisition system stimulated by dim stars, secondly, we propose and evaluate an algorithm designed to operate at high speed and to be compatible with an Field-Programmable Gate Array implementation, before evaluating the performance of the implementation on FPGA. Finally, we debate the usability of such a system, both in terms of compatibility with a mission and CubeSat ecosystems, and in terms of performance. As a result, aside from removing the need for a rate gyro, Attitude Determination and Control Systems overall pointing performances can be increased. The proposed attitude determination system achieved a 0.001° accuracy, with a 99.1% sky coverage and an ability to reject false-positive while performing a single-frame lost-in-space star identification at a 50 Hz update rate with a total delay of 19 ms, including 13 ms.

## 1. Introduction

With the development of the satellite industry, the requirement of commercial-off-the-shelf (COTS) subsystems is rapidly increasing. As it needs to be designed for each mission, Attitude Determination and Control Systems (ADCSs) have been resisting this trend, limiting the development of COTS to actuators and single sensors. Attitude Determination has greatly profited from the use of star trackers [1,2,3,4]. However, as star trackers are intended to be the most accurate Attitude Determination sensor, these are required to use a relatively long exposure time to gather more light and, therefore, increase the number of detected stars and better estimate their position. This low update rate has been traditionally compensated through the use of a rate gyro in most satellites, ranging from Hubble Space Telescope [5] to microsatellites [6]. As a result, satellite designers are required to invest more resources in developing this crucial subsystem.

Moreover, a mission hosting various payloads, or payloads required to successively point at various attitude will require wide-angle maneuvers with high angular rate, blurring the images acquired by a star tracker and temporarily disabling this crucial sensor, relegating the attitude determination system to a drifting integrator, subject to drift during the duration of the operation.

On another hand, increasing the update rate of available star trackers in a range between 25 and 100 Hz and reducing the latency would solve most of those problems. The main benefit being the capability to perform large-angle maneuvers with direct feedback control from the optical input and providing a drift-free attitude measurement at the expense of accuracy, as the high-update rate requires a short exposure time, ignoring limiting the number of stars that can be perceived. The existence of such a star tracker would benefit the ecosystem by proposing a different tradeoff to configure an ADCS suitable for the mission’s requirements.

Increasing the speed of star trackers has been a common research subjects in recent years, Jiang et al. [7] achieved a 10 Hz update-rate as early as 2009, using an AT91 platform and a Field of View (FOV) of 12°. More recently, Wei et al. [8] from the same laboratory have been investigating variable-rate gyroscope for a given angular rate in order to provide faster reaction time at high angular velocity while providing a more accurate output for fine adjustments.

A recent breakthrough being StarGyro100 [9] which proposes a novel approach implementing a 100 Hz image-based gyroscope, combined with a 100 ms lost-in-space search to provide a 100 Hz update-rate.

This approach has the added benefits to demonstrate the possibility to perform multiple measurements from a single image sensor, which is particularly suitable on FPGA platforms where the video stream could be duplicated and fed to multiple instrumentations Intellectual Properties blocks (IP blocks). This way, a single sensor could be used as a star tracker, a rate gyroscope, a sun sensor, and a horizon sensor, and perhapse allow ADCS to only rely on optical measurements [10].

This work extends Marin 2018 [11] and explores the feasibility of high-speed star-tracker performing lost-in-space attitude determination on a single frame at the highest speed granting a single sensor solution to Attitude Determination which could both increase the overall pointing performances of ADCS, raise the maximum angular rate, and reduce the design efforts of future spacecrafts, or used besides MEMS sensors to increase the overall robustness of the ADCS system. This work only estimates the attitude of the satellite, and not it’s rate. A mission relying on this work can estimate the angular velocity of the satellite’s body using [12,13], or continue using an IMU sensor for added precision and redundancy.

This paper first provides a preliminary analysis of the requirements for a high-speed star tracker in Section 2. We then describe and validate the proposed star tracker in Section 3 before presenting the architecture and performances of its implementation on Field Programmable Gate Array (FPGA) in Section 4. Finally, future work and conclusions are provided in Section 5.

## 2. Feasibility of Short Exposure Time Star Trackers

### 2.1. Hipparcos Star Catalog

The most reliable knowledge of distant stars lies in various star catalogs, which have been constructed during the dedicated satellite missions. This work relies on the Hipparcos Star Catalog [14] which provides the equatorial coordinates, apparent magnitude, and proper motion of 118,218 stars with an apparent magnitude of 9 and lower.

### 2.2. Analysis of Star Luminosity and Feasibility of Short Exposure

Before studying the exposure time of the sensor, the necessary FOV to have a sky coverage of 99% must be studied. Figure 1 plots the percentage of the sky containing *N* or more stars with a given apparent magnitude of less. Figure 1a,b show that achieving a 99% sky coverage while only using stars with an apparent magnitude of less than 3 or 4 would require a FOV greater than 90°, which is unreasonable. Figure 1c,d show that limiting the apparent magnitude to 5 or 6 respectively requires a FOV of 50° and 35°. Fainter stars were not considered in this study.

As the high update rate of the proposed star tracker requires a shorter exposure time, the proposed star identification algorithm’s design, analysis and implementation will be built upon the 1448 brightest stars whose apparent magnitude is lower than 5.

### 2.3. Constrains on Optics and Exposure Time

Before any further development, the feasibility of a high-speed star trackers must be studied. This section provides a preliminary study quantifies the sensibility a sensor and its optics given the spectral flux density of α-Lyrae, and extrapolate its results to estimate the sensibility at various apparent magnitude.

The flux density $F_{\lambda }$ of a star for a given wavelength λ and bandwidth BW can be converted into a photon count $F_{\gamma }$ using the Planck-Einstein Relation:(1)$$E_{\gamma }=\frac{c\cdot h}{\lambda },$$
and
(2)$$F_{\gamma }=\frac{F_{\lambda }\times BW}{E_{\gamma }}.$$

The photon flux on the sensor can then be computed with regard to the lens’ Aperture $A_{lens}$ and Transmission $T_{lens}$:(3)$$F_{\gamma /sensor}=F_{\gamma }\cdot T_{lens}\cdot \pi \cdot {\left(\frac{A_{lens}}{2}\right)}^{2},$$
and it’s associated electron flux as a function of the sensor’s Quantum efficiency $Q_{e}$:(4)$$F_{e^{-}/sensor}=Q_{e}\cdot F_{\gamma /sensor}.$$

And finally, we obtain the total number of accumulated electrons over a period *T* on the sensor:(5)$$N_{e^{-}/sensor}=F_{e^{-}/sensor}\cdot T.$$

Hayes 1985 [15] provides us with reference spectral data of α-Lyrae allowing us to compute the flux of photons from a α-Lyrae with an apparent magnitude of 0.03 emitting a flux of $F_{\lambda }=3.44\times 10^{-8}\;Wm^{2}\mu m^{-1}$ in the V band centered on $\lambda_{V}=555.6\;\mathrm{nm}$.

Considering the definition of the apparent magnitude V defined in Johnson’s UBV photometric system [16] and used by [14,15], we can extrapolate the number of electrons per pixels to any star with a from its apparent magnitude *V* by considering a similar specter in the V band as follow:(6)$$N_{e^{-}/sensor}\left(V\right)=N_{e^{-}/sensor}\left(V_{\alpha -\mathrm{Lyrae}}\right)\times 10^{-(V-V_{\alpha -\mathrm{Lyrae}})/2.5}.$$

Finally, the distribution of light on each individual pixel can be modeled using a normal distribution with a standard deviation σ as the Point Spread Function (PSF). The individual pixel values is then computed according to the pixel’s position, the star’s magnitude $V_{s}$ and its center position on the sensor ($x_{s}$, $y_{s}$):(7)$$N_{e^{-}/pixel}\left(x,y\right)=N_{e^{-}/sensor}\left(V_{s}\right)\cdot \frac{1}{\sigma \sqrt{2\pi }}\cdot \exp\left(-\frac{1}{2}\cdot \frac{{x-x_{s}}^{2}+{y-y_{s}}^{2}}{\sigma^{2}}\right).$$

The CMOS sensor Onyx from Teledyne was selected for the first demonstrator for its low illumination performances, high configurability and 14-bit readout. In addition to the sensor’s characteristics, its reference kit uses a reprogrammable Artix-7 FPGA as its sensor front-end. This allows the use of a design methodology that is compatible with some CubeSat’s Onboard Computer (OBC). Table 1 summarize the optical parameters of our first prototyping platform with two different optics.

We can obtain in Table 2 an estimation of the electron stimulation of the sensor’s pixels for two different optics, two different focus and three stars chosen among the Hipparcos catalog [14]. The results in Table 2 predicts average center pixel value when stimulated by the light of three given star for an exposure period of 20 ms for the two evaluated optics. The dimmest evaluated star λ-Lyrae (V = +4.94) corresponds to the dimmest stars the proposed star tracker operates with. We expect a 14-bit read-out value of 944 and 436 for λ-Lyrae if the optics follows the PSF modeled with a standard deviation of one pixel.

This preliminary study concludes that it is possible to detect stars up to an apparent magnitude of 5 at 50 frames/s with a large enough, but reasonable optics. Moreover, with a 14-bit sensor, it is possible to choose an aperture that both prevents over-exposure with stars of an apparent magnitude of 0 while still granting enough sensitivity to detect stars with an apparent magnitude of 5. In addition, similar computations show that a FOV of 36° would be acceptable and allow the use of stars with an apparent magnitude of 6. This research will proceed with a 50° FOV and stars with an apparent magnitude below 5 for simulation convenience, but both of the 36° and 51.6° commercially available lens shall be tested in the field at a later stage, and chosen according to their performance overall performances, including Noise Equivalent Angle.

## 3. Proposed Algorithm

The proposed star identification algorithm relies on the feature and feature-descriptor [17] concepts commonly used in Computer Vision in order to produce a rotation-invariant feature-descriptor vector that could be both be computed and identified efficiently on the target platform.

Many star identification algorithms have been proposed, among the most popular, we find the Triangle Algorithm, the Grid Algorithm and the Match Group Algorithm as stated in [18,19]. Those algorithms often rely on matching many ambiguous features descriptor matches and deciding the star identity through consensus. This makes star identification more robust, but also more computationally and memory demanding, which complexify their real-time and fault-tolerant implementation on a FPGA.

The Pyramid algorithm [3] presents an interesting approach with the ability to determine the attitude from only four stars, especially suitable for high-speed star trackers with a short exposure time. However, the authors believe that the iterative approach of this algorithm would be detrimental to an efficient implementation under the given constraints. We propose to identify similar star clusters of fixed size to construct a feature descriptor that could be matched without ambiguity against a database built upon the Hipparcos Star Catalog. The said feature descriptor was specially tailored to be easily implemented on FPGA with a low resource usage while achieving the targeted update rate with a lost-in-space attitude determination on every single frame.

### 3.1. Feature and Feature Descriptor

While a high accuracy star tracker collects as much light as possible, a high update rate star tracker will only have access to the brightest stars and focuses on producing a real-time output.

With *N* stars we are presented with 2N star spot centroid position parameters. Among those, we can extract 3 parameters for the cluster’s location and angle on the frame and 2N−1 parameters to compose the feature descriptor:$\delta_{1,\ldots ,N}$, the arc lengths between each star and the mean position of the stars,$\alpha_{1,\ldots ,N-1}$, the angles between each star, the mean position of the stars, and the furthest star to this mean position.

The concatenation of those two vectors constitute the feature descriptor $FD=[\delta_{1},\ldots ,\delta_{N},\alpha_{1},\ldots ,\alpha_{N-1}]$ and the feature location information consists of (x,y) the mean position of the stars, and $\alpha_{0}$ the angle between the horizontal and the line passing through the mean position of the feature, and the furthest star. Every angle is referenced with the furthest star to the mean position for added accuracy in the estimation of the sensor’s roll angle. Figure 2 shows an example of a feature descriptor with four stars.

When a feature is composed of less than *N* stars, the remaining arcs and angles are zeroed. This ensures that a feature composed of more stars cannot be matched with features composed of fewer stars, allowing a seamless handling of frames with a low star count without loss of accuracy. This way, we can identify frames with a high star count with 9 feature descriptor parameters, and frames with only three stars with 5 feature descriptor parameters. Experiences show a feature built upon maximum N=5 stars to work best.

### 3.2. Feature Database

The feature database is constructed after the brightest feature from every possible frame. The duplicated and non-unique features are filtered from the database.

The feature database containing every existing feature contains 2584 entries for a total size of 77.5 kB. Each entry is composed of the feature descriptor, nine 16-bit floating-point numbers, and the feature’s attitude, three 32-bit floating-point numbers. As shown in Figure 3, those features are homogenously distributed on the celestial sphere.

### 3.3. Star Detection

As stars can be considered at infinity, the optics projects each star to a single point on the imaging sensor if perfectly focused. However, the optics can be adjusted in order to obtain a PSF spanning over multiple pixels, and achieve a sub-pixel detection of the star centroid, increasing the overall accuracy of the proposed star tracker.

For an efficient hardware implementation, the star detection relies on a sliding detection window. A first 3×3 detection window is sliding on the frame in order to detect every star. Then a preliminary rejection of twin stars is performed as those would appear on a single pixel on a sensor.

Once the stars have been detected, the luminance and centroid of the spot are computed on a Wsize×Wsize pixels patch. This allows a more accurate detection of the luminosity along with sub-pixel detection of the position of the star. The size of the integration window Wsize is to be chosen during the implementation. The effect of Wsize is debated in the next section.

### 3.4. Star Identification and Feature Descriptor Matching

As this algorithm relies on feature descriptors, the search for a match with the stars feature descriptor $\mathrm{featdesc}\;\{s_{1,\ldots ,N}\}$ is reduced to that of the closest feature descriptor from the database $\left\{{\mathrm{FD}}_{i}\right\}$ as in Equation (Equation 8) with a score *m*.
(8)$$m=\min_{i}\{\mathrm{dist}({\mathrm{FD}}_{i},\mathrm{featdesc}\;\left\{s_{1..N}\right\})\}$$

The distance dist being a Manhattan distance modified to account for each quadrant of the angle’s differences such as:(9)$$\mathrm{dist}\left(\left[\delta_{1,1},\ldots ,\delta_{1,N},\alpha_{1,1},\ldots ,\alpha_{1,N-1}\right],\left[\delta_{2,1},\ldots ,\delta_{2,N},\alpha_{2,1},\ldots ,\alpha_{2,N-1}\right]\right)=\sum_{i=1}^{N}\Delta \delta_{i}+\sum_{i=1}^{N-1}\Delta \alpha_{i}$$
where: $\Delta \delta_{i}=\left|\delta_{1,i}-\delta_{2,i}\right|$ and $\Delta \alpha_{i}=\min(|\alpha_{1,i}-\alpha_{2,i}\left|\;\left(\mathrm{mod}\;2\pi \right),2\pi -\right|\alpha_{1,i}-\alpha_{2,i}|\;\left(\mathrm{mod}\;2\pi \right))$.

In order to improve the robustness of the algorithm, two improvements were explored: increasing the feature descriptor’s number of parameters, and testing subset of stars whenever more stars are available. The former strategy was found to decrease the stability of the algorithm as it does not increase the matching accuracy, but dramatically increases the sensibility to undetected and wrongly detected stars. The latter does increase the stability by allowing the algorithm to pick the best feature among the ones available for a given set of stars. This strategy has proven to be effective, but increases the computation time of the feature matching algorithm by a factor N^N where $\hat{N}$ is the number of stars in the set to explore and *N* is the number of stars necessary to build a feature. A good compromise between robustness and performance can be achieved with $\hat{N}=N+2$. Thus, Equation (Equation 10) operates on every subset of *N* stars among the N+2 brightest stars as described in Equation (Equation 8). When not enough stars are available, the additional stars are replaced with zero-magnitude stars as a guard value, which is later converted to a zero-padded feature descriptor, and ignored during feature matching.
(10)$$m=\min_{i}\min_{j,k\in [[1..N+2]]}\{\mathrm{dist}({\mathrm{FD}}_{i},\mathrm{featdesc}\;\left\{s_{\neq j,k}\right\})\}$$

The algorithm returns N+2N feature locations on the frame and their associated matching score. When the matching score *m* is higher than a threshold value, the star identification must be considered as defective, and the estimated attitude should be discarded as detailed in the next section.

### 3.5. Validation

This section validates the performances of the algorithm’s software implementation. We first analyze the performances and noise robustness of the feature descriptor matching algorithm with noisy star position and magnitude information. Secondly, the overall accuracy is evaluated against randomly generated pictures, finally, we provide a simulation estimating the expected pointing accuracy obtained by replacing a classic ADCS using a low-speed Star Tracker and a Rate Gyroscope with an ADCS using the proposed star tracker as its sole attitude determination sensor.

#### 3.5.1. Validation of the Feature Matching Algorithm

Figure 4 shows the output error of the star tracker when each star’s position is injected with a 0.1° and 0.2° noise. It exposes the sensibility of the algorithm to noise. The graphs Figure 4a,b show the presence of erratic star identification. However, as graphs Figure 4b,e show, the matching score does degrade and provide an insight of this error and allows the star tracker to reject the uncertain estimations whenever this score is above a threshold. Graphs Figure 4c,f show the error of filtered (score < 1) results to be lower than the input noise.

Table 3 presents the filtered and unfiltered performances of the star matching algorithm when subjected to different noise models. The first line of Table 3 shows the perfect accuracy of the star identification algorithm in noiseless conditions. This validates the star feature descriptor and its matching algorithm and proves the consistency of the database.

Lines 2 and 3 of Table 3 provide a quantification of the improvement achieved by filtrating unconfident star identifications. The Root Mean Square Error (RMSE), initially of 3.699° and 3.386° on 2500 samples, decreases to 0.039° and 0.076° by rejecting the 0.08% and 0.27% unconfident estimations.

A variation in the stars’ position corresponds primarily to each star’s proper motion. Evaluating this sensitivity allows the determination of the duration over which the star feature database can be considered as reliable for the required Attitude Determination accuracy. Over time, the mean output error of the algorithms increases by about a third of the amplitude of the variation of the star position. Considering roughly that most of the stars have a proper motion lesser than 1000 ms of arc per year, this leads to a decrease of the star tracker accuracy of 0.00028° variation per year. This yields to a database validity of 5 years for a 0.001° accuracy.

Finally, the last two lines of Table 3 show no sensitivity to variations of the apparent magnitude by ±1 (+150%, −60% light intensity). This validates the robustness of the sensor to variations of intensity, which may occur on two occasions:Dead pixels will appear over time, due to the exposure to radiationsShot noise will induce a noise of magnitude $\sqrt{N_{e^{-}}}$.

This test demonstrates the robustness of the proposed algorithm to those two light intensity noise sources. Additional plots completing Figure 4 with every noise model shown in Table 3 if available in Appendix A.

#### 3.5.2. Star Identification from a Celestial Image

To produce test-frames, various projection models can be used. Both pinhole-model projection and planisphere projection were studied. Although a pinhole-model projection would be most accurate, and include the optical model of the camera, it would require the generation of a prohibitively large dataset. As the proposed method does not rely on a given optics, a planisphere projection was chosen, in order to profit from the ability to generate every test-frames with the same roll angle from a single planisphere image of the sky. A planisphere of the sky corresponding to the star tracker’s Field of View of 50° and horizontal resolution of 1000 pixels. This results in a 7200×3600 pixels image of the sky composed of every star up to an apparent magnitude of 7. One of the design parameters of a star tracker is having the lens slightly out of focus, to project each star’s image on a wider area on the image sensor and produce larger spots. This is necessary in order to achieve sub-pixel detection of the star position. Similarily as with Equation (Equation 7), the test sky image models each stars’ PSF with a normal distribution of standard deviation σ as defined in in Equation (Equation 11). Similarily as Equation (Equation 6), the intensity *I* of each pixel (x,y) are computed from each star *s*’ apparent magnitude $V_{s}$ and its Right Ascension $x_{s}$ and Declination $y_{s}$ in pixels as follow:(11)$$I\left(x,y\right)=\sum_{s}10^{-V_{s}/2.5}\exp(-\frac{{\left(x-x_{s}\right)}^{2}+{\left(y-y_{s}\right)}^{2}}{\sigma^{2}}).$$

Table 4 shows a summary of the star identification accuracy of the proposed star tracker for different spot size σ and integration window size Wsize. For this test, the star tracker operates on cropped frames of 1000 × 1000 pixels from this reference sky image and tested 10,000 times at a random attitude for different spot width and window size Wsize.

As shown in Table 4, the RMSE of the simulated star tracker running on an image of the celestial sphere. Those results show that a minimum PSF standard deviation $\sigma^{2}$ of 1.0 pixels is necessary in order to achieve a sky coverage above 98.8%, and sub-pixel detection within a 0.001° accuracy. Although a PSF with a standard deviation above 1.0 pixels is not necessary to achieve sub-pixel accuracy, it does yield better sky coverage. We observe no influence of the size of the detection window on the accuracy. However, it is expected to improve the accuracy of the detection of the centroid as well as improve robustness to shot noise and motion blur.

The python implementation of the algorithm runs in 15 ms on a single thread running on a 3.8 GHz CPU. A real-time demonstration video has been provided with the submission of this paper.

#### 3.5.3. Influence of Noise Sources

The following section evaluates the effect of different noise sources. Table 5 provide a qualitative estimation of the effect of various noises, their synergies with the proposed star tracker’s design parameters, quantify their impact on the pixel data, and finally estimate the overall impact on the system if those noise sources cannot be mitigated. In the case of noise sources producing a centroid position noise or a magnitude noise, we can expect an acceptable performance according to Table 3. The most challenging noise sources are angular rates above the estimated limit, global blooming when the sun is in the Field of View, as well as the shot noise as those sources of noise would induce false star detections and missed star detections. In such an occurrence, the proposed algorithm will evaluate different subsets, and return an erroneous results if it fails to identify the star cluster. If the algorithm is not able to find a proper match, the associated matching score would most likely rise above the rejection threshold, preventing any false identification.

The maximum angular rate can be estimated though the use of a motion blur corresponding to angular speed. We simulated such noise by averaging 15 frames homogenously distributed along the attitude path for various given angular rate and different Wsize. Each test were performed on the same 10,000 random attitudes. Each test uses a diagonal motion. For each test, Table 6 provides us with percentage of confident results, and their associated RMSE.

We observe that the star tracker’s sky coverage drops at 15°/s for Wsize=3, 20°/s for Wsize=5 and 30°/s for Wsize=7 and Wsize=9. We can safely assume a 15°/s maximum angular rate for Ksize=7, which increase the RMSE by 50%.

Another major source of noise and loss of availability is the passage of the Sun, Moon and Earth in the Field of View. Future work will need to evaluate the noise and loss of availability precisely before deployment. Figure 4 of [20] provided us with design guidelines for a a baffle and its automatic vane design. The provided equation shows the relation between the baffle’s length *L*, the lens’ half aperture *a*, the FOV α and the Earth’s Aspect Angle $\phi_{T}$:(12)$$L=\frac{2\cdot a}{\tan\left(\phi_{T}\right)-\tan\left(\alpha \right)}.$$

Table 7 provides the estimated baffle length using Equation (Equation 12) on four optical systems. The first two lines corresponds to the same lens as in Table 1, the last two lines corresponds to lens with a slightly smaller aperture, in order to reduce the baffle size. The actual baffle size must be chosen according to the mission’s constrains.

#### 3.5.4. Case Study for Gyroless Applications

This section presents a simple comparison of the performances of a reference system using the true attitude, a test ADCS using only the proposed star tracker operating at 25 Hz using a fading-memory filter (G=0.5) [21] to improve the stability of the estimation of the angular speed, and a concurrent ADCS using a 2 Hz star tracker and a 50 Hz interpolation using a MEMS rate gyroscope. This MEMS rate gyro has a noise of 0.1°/s, comparable to the best MEMS gyro on the market. As this work focuses on the development of a sensor, this simulation does not implement any additional filtering.

The controlled system consists of a satellite with a moment of inertia Diag(2,1,1), controlled with a pure torque. The feedback control system consists of a PD controller (Kp=0.2,Kd=1.0) operating with the last 3 estimated attitudes. Figure 5 shows the unit response to a command of attitude $\vec{\gamma }=\left(-120{}^{\circ},0,0\right)$ and angular speed $\vec{\omega }=\left(0,0,0\right)$.

All three systems presents the same global convergence as can be observed in Figure 5a,d,g. Figure 5’s third column shows the accuracy of the ADCS once it has settled. As expected, the error of the control system with the true attitude continue to converge toward zero. The ADCS system using a 2 Hz star tracker and a 50 Hz gyroscope manages to bound its pointing accuracy within 0.084° after 45 s, but Figure 5c exhibit a noisy output. Finally, the ADCS system using the proposed star tracker at 25 Hz operating on simulated images presents a more stable output on Figure 5i and converges toward a lesser error, bound to 0.0012° after 45 s. In both cases, the result could be improved using various filtering methods, sensor fusion and other control technics.

Those results validate the use of the proposed high update rate star tracker in order to reduce the Bill of Material of satellites while improving the accuracy of the ADCS.

## 4. Implementation of Star Tracking Algorithm on Final Platform

### 4.1. Ecosystem

As the proposed star tracker is part of a CubeSat mission, it must be implemented on a flight computer already available in the CubeSat ecosystem. Standalone FPGA are rare on CubeSat. However, with the rise of FPGA Systems on Chip integrating a processor, and the recent trend to evaluate commercial-grade component for non-critical components of space missions, NASA NEPP [22] have been evaluating Zynq and MAX10 SoC’s tolerance to Total Ionizing Dose (TID) dose as well as Single-Event Upset (SEU) and Single-Event Latch-up (SEL) tolerance. In addition, various CubeSat OBC featuring a Zynq XC7Z020 have been commercialized. This work focuses on NanoMind Z7000 OBC from GOMspace, and uses Vivado HLS for the implementation of the star tracker. Space grade solutions would include Microsemi’s lineups of radiation-tolerant FPGA as well as Xilinx’s fault-tolerant Virtex-5 or Kintex Ultrascale FPGA.

### 4.2. Architecture

The implementation of the star tracker follows the image processing pipeline described in Figure 6 and communicates using AXI4-Stream [23] interfaces with input and output blocks, as well as between sub-blocks. Every AXI4-stream interface encodes fixed-decimal integers, signed or unsigned, as well a AXI4-Stream’s tlast side-channel, which is used as an end-of-frame bit, requesting a pipeline flush, as well as the computation of the star features once the end of the frame has been reached.

The different processing stages operates in the following sequence:The first stage extract_ROI consist of a 3×3 detection window, which forwards the 3×3 pixel patch to the next block when a maximum is detected. It takes an 8-bit grayscale image input at 100 MHz, and outputs nine 8-bit pixels as well as two 12-bit window coordinates on a 96-bit bus at a lower data-rate. This IP is fully pipelined with an initialization interval of 1 cycle, allowing the maximum throughput.An AXI-4 Stream FIFO provides a buffer to prevent pipeline stalls on the previous block, when the next block performs a non-pipelined computation.The star_detector block computes the star’s statistics, such as sub-pixel centroid position, apparent magnitude and possibly a filter to reject star detections if not fitting a model. It outputs three 16-bit unsigned fixed-decimal representing the detected stars’ position and apparent magnitude on a 46-bit bus. As this block’s input has a low data rate, it is not pipelined and performs a blocking operation lasting 174 cycles in order to reduce its resource usage. This block should implement a conversion from a pinhole-model projected input image to angles (equivalent to planisphere projection) if a pinhole-model projection test image or a camera is used. Any other distortion correction can be implemented here.The feature extractor operates in two stages:–A live bubble-sort shirt-register sorts the N+2=7 brightest stars as they arrive.–Upon receiving an end-of-frame signal (tlast), a features position and feature descriptor is computed from each the N+2N permutation of stars, and the data is sent on a 192-bit bus. This IP relies on two CODRIC functions: atan2 and hypot.Finally, the feature matcher takes a second AXI4-Stream for the feature database, and perform a search in the database for each 21 features’ closest match, and latches each feature’s position on the screen and the reference position of the identified feature on four 24-bit numbers as well as a 16-bit estimation of the orientation of the feature and 16-bit for the matching score. Upon reception of the database’s tlast signal, the 21 feature’s closest matches are sent on the 128-bit output bus.

An additional block to compute the attitude could be added, but low-performance floating-point operations are more efficiently performed on a microcontroller or microprocessor.

#### 4.2.1. Evaluation of the Throughput and Implementation Weight

The implementation is designed to work at the pixel clock of 100 MHz, which would provide a 76 Hz update rate. Real numbers would be slightly lower due to the transfer not being continuous, but should properly operate at a maximum update rate of 50–60 Hz. Section 4.3 tested the IP at a frequency of 167 MHz without any glitches, and was able to transfer a frame every 8.93 ms, even though the tests ran at 51 Hz due image copy and cache synchronization on the processor side. The induced delay by the star tracker is of maximum of 158,581 clock cycles (1.59 ms at 100 MHz) for a database containing 2176 elements. Pipelines stalls are more likely to happens between the ExtractROI and the StarDetector block, hence the FIFO in between. The FIFO’s depth can be adjusted as required. When accounting for the image’s transfer time (1280 × 1024 clock cycles, 13.1 ms at 100 MHz), the total delay from the image’s capture to the attitude determination is of 14.7 ms.

Total estimated power usage by Vivado is of 1.372 W, of which 1.262 W is attributed to the processor, and 0.110 W are attributed to the FPGA design, including the testing elements.

Finally, the resource usage on the FPGA is an important metric to quantify the space required for the proposed star tracker. Digital Signal Processing blocks (DSP) and Block-Ram (BRAM) consumption are low, while logic usage account for a total of 16.8% of the FPGA, details are provided in Table 8. feature_extractor_st_0’s high slice usage is related to the usage of hls::hypot and hls::atan2 which uses the CORDIC algorithm to compute the associated functions, while feature_matcher_strm_0’s high logic usage could probably be lowered by using a DSP for the computation of the feature’s matching score.

#### 4.2.2. Test Setup

For convenience, the validation was performed on a Pynq-Z1 platform, allowing the use of Python for FPGA control, Direct Memory Access (DMA) transfer control, as well as data analysis and plotting. A block design was created as detailed in Figure 7 and in additional block diagrams from Appendix B, where two DMA are providing the input image and input database to the star tracker, and an output block stores the 21-stars feature matches into a block memory whose secondary port is directly accessible from the main processor. The test protocol uses the same celestial image as Section 3.5.2, and consist of two phases: building the feature database, then using this database to estimate the attitude of random frames.

### 4.3. Evaluation of the Performances of the Implementation

#### 4.3.1. Database Adjustments

In order to compensate for discrepancies between the python simulation and fixed-decimal implementation on FPGA, a new database must be generated from the feature extracted by the FPGA implementation. To do so, a special bitstream was created without the feature-matching IP, in order to recover the feature extraction. Using the reference unrotated celestial image, a feature extraction has been performed for every degree of Right Ascension and Declination, the *k* brightest feature was inserted in the database. This resulted in the construction of a database whose statistics are provided in Table 9. The choice of the size influences the total computation delay, but does not impact the update rate of the sensor as long as the delay does not induce pipeline stalls. Table 9 shows an increase of sky coverage as *k* increase. k=5 is a good tradeoff with a high sky coverage, good RMSE and 21 ms of delay without reduction of throughput. The associated database will be used for further validation.

#### 4.3.2. Evaluation of Attitude Determination Accuracy without Rotations

Tested over 1 million random frames selected from the reference sky image used in Section 3.5.2, the tested star tracker obtained a 99.8% overall success rate (1534 frames with error above 0.02°).

#### 4.3.3. Evaluation of Attitude Determination Accuracy with Rotations

A similar protocol is applied for every 5° of rotations of the celestial sphere. However, as the rotations caused artefacts on the edges, the tests were limited to (RA∈[0–300°] and DE∈[0–130°]). Figure 8a–d presents the True-Positive, True-Negative, False-Positive and False-Negative at any roll angle, considering that a positive result is an attitude identification within 0.02° (one pixel). Figure 8a shows a sky coverage (True-Positive) of 97.7% for this specific implementation, hinting the necessity to improve the performance of the star tracker, though the use of an expended database, or refining of the algorithm itself. Figure 8c is coherent with previous results, as the algorithm’s very strong false-positive rejection. We observe a rate of False-Negative of 0.0096%, slightly higher than the expected rate of randomly landing within 0.02°. Finally, Figure 8c shows the mean RMSE of the confident results for every roll angle to be 0.00099°. In some extremely rare occurrences, false-positive occurred when some star configuration were mapped onto a features from the database. In such an event, removing the faulty feature from the database allows the system to recover from this error. Experiments showed no loss of sky coverage or accuracy. As this situation isvery hard to reproduce, it is impossible to give precise numbers, but such occurence has happened in less than 100 part per billions, ensuring an excellent rejection of false-positive.

Figure 9a shows the 2D histogram of the repartition of errors, regardless of the error’s magnitude, and Figure 9b provide a 2D histogram of the repartition of stars with an apparent magnitude higher than 5 for comparison. We observe an increase of errors for attitudes pointing outside the galaxy, as those attitudes present the lowest number of stars. Figure 9c,d provide the histogram of error and unconfidence for reference.

#### 4.3.4. Modification for a Mission

This work focuses on the design and validation at different stages of the development of a high-speed star tracker, however, it is important to discuss the integration of such a system within a mission. As with the rest of this work, this approach relies on a 7-series FPGA from Xilinx, which is not compatible with Microsemi’s space-grade FPGA, and a re-implementation would be necessary. However, this implementation is can be embedded on Xilinx’s new XQRKU060 space-grade FPGA, as well as many Zynq-based CubeSat OBC without modification of the star tracker’s IP itself.

Figure 10, Figure 11 and Figure 12 presents integration schemes to integrate the proposed star tracker for a mission.

Figure 10 shows how the star tracker’s input block can be connected to a camera regardless of its interface (LVDS, MIPI-CSI, Parallel, GigE-Vision, …). It is important to note that the camera interface block could perform any preprocessing, such as input width reduction from 14 to 8 bits, with a Gamma correction in order to keep the dynamic range. Extra filtering such as a median-filter to reduce shot noise might also be relevant.

While Figure 10 keeps the Zynq processor, as this processor has been used in various CubeSat missions embedding a NanoMind Z-7000 OBC, it is important to keep in mind that a COTS Star Tracker would be separated from the OBC, and would output the data on a CAN or RS-485 interface. Figure 11 demonstrates how a Microblaze could provide such an interface. A Microblaze could also serve other purposes, such as configuring the camera.

Finally, Figure 12 shows a Triple Modular Redundancy (TMR) implementation of the given star tracker, using Vivado’s newly introduced TMR architecture blocks. This option might be interesting for critical subsystems if the whole ADCS subsystem must be fault-tolerant. On Xilinx’s technology, a TMR Microblaze soft-core processor is available, as well as Soft-Error Mitigation of the configuration memory for Ultrascale series. Leveraging those IP could provide a completely fault-tolerant solution, for application where fault-tolerance is required, but not high radiation tolerance.

## 5. Conclusions

This work described the design and validation of high-speed star trackers at many stages of the development. It demonstrated the following points:The feasibility of a 25–50 Hz star tracker with a 50° FOV exploiting stars down to a magnitude of 5 while offering a 99% sky coverage. Such detections equates to 468 ${\mathrm{LSB}}_{14}$ on the evaluated sensor and optics with a 20 ms exposure.The proposed star matching algorithm presents a proper noise robustness, a very strong False-Positive rejection, and estimates the attitude with a 0.001° RMSE with a 99.2% sky coverage when limited to a magnitude of 5.0.The algorithm can be implemented on a CubeSat OBC and perform a lost-in-space attitude determination at 50 Hz, with a 21.0 ms delay presenting a sky coverage of 97.7% with an RMSE of 0.001° on a noiseless dataset, while using 17% of available logic.

As a comparison, Table 10 provides a comparison between the expected performances of the proposed star tracker.

Considering those results, the authors will pursue investigations with pinhole-model datasets, ground tests and embedding in a CubeSat mission. Such a star tracker could increase the overall pointing accuracy and stability of satellites for low-cost missions, by allowing a direct control from the camera sources. Other sensors might still be used to increase performance or redundancy.

## Figures and Tables

**Figure 1 sensors-20-02388-f001:**
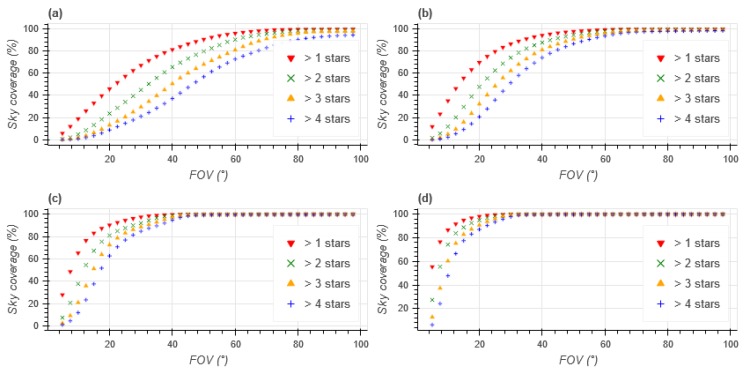
Sky coverage vs FOV for given maximum apparent magnitude (**a**) mag < 3 (**b**) mag < 4 (**c**) mag < 5 (**d**) mag < 6.

**Figure 2 sensors-20-02388-f002:**
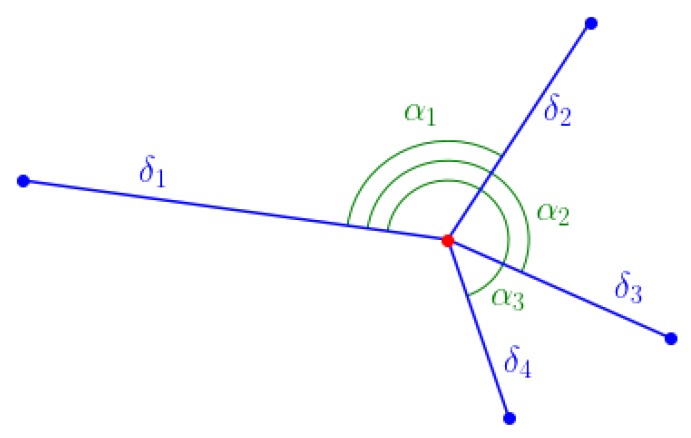
Feature descriptor of four stars.

**Figure 3 sensors-20-02388-f003:**
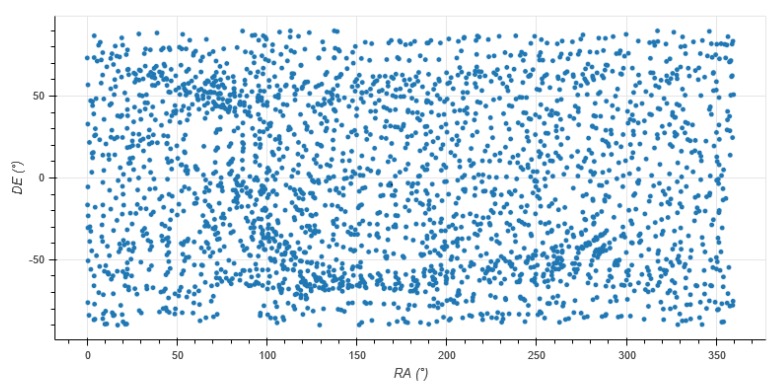
Distribution of star features in the database.

**Figure 4 sensors-20-02388-f004:**
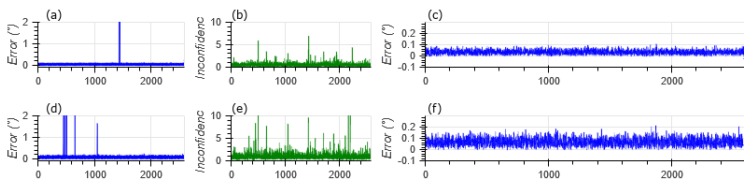
Star catalog to feature database matching with angular noise: (**a**) Raw Error with ±0.1 Noise (**b**) Matching score (±0.1 Noise) (**c**) Error after filtering (±0.1 Noise) (**d**) Raw Error (±0.2 Noise) (**e**) Matching score (±0.2 Noise) Noise (**f**) Error after filtering (±0.2 Noise).

**Figure 5 sensors-20-02388-f005:**
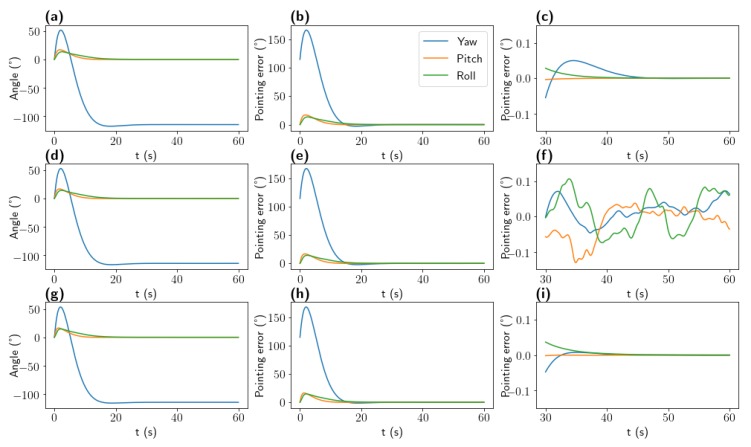
Pointing accuracy performances of three ADCS; Top (**a**–**c**): ADCS with ground truth attitude, Center (**d**–**f**): ADCS with 2 Hz Star Tracker and 50 Hz rate gyro Bottom (**g**–**i**): ADCS with the proposed Star Tracker, Left (**a**,**d**,**g**) Impulse response, Middle: (**b**,**e**,**h**) Pointing error: (**c**,**f**,**i**) focus on the pointing errors after 30 s.

**Figure 6 sensors-20-02388-f006:**
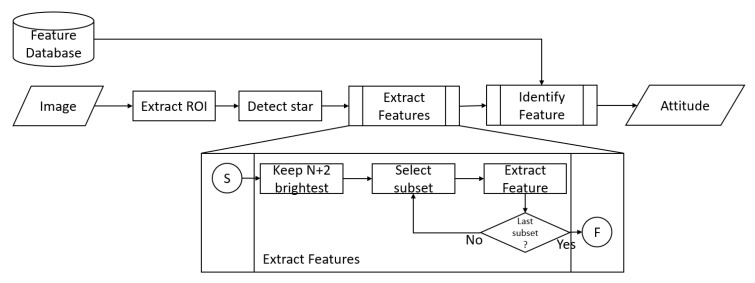
Flowchart of the proposed Star Tracker image processing pipeline.

**Figure 7 sensors-20-02388-f007:**
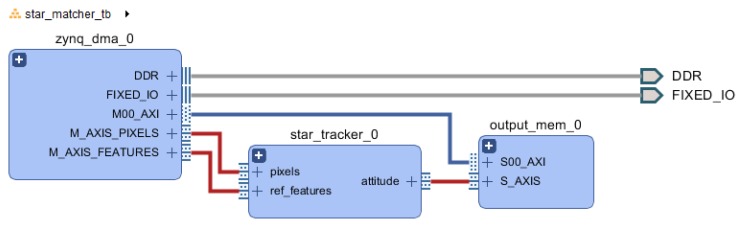
Top level design of test setup with Zynq.

**Figure 8 sensors-20-02388-f008:**
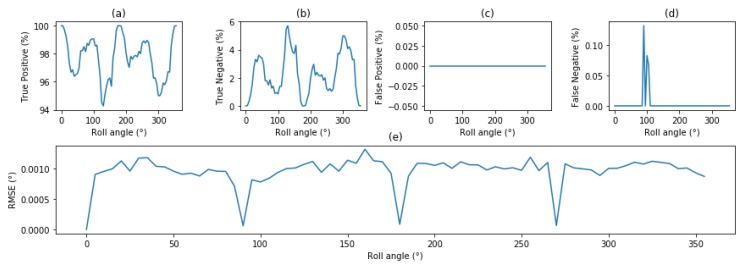
Evaluation of the performances of the proposed star tracker on 10,000 random images for every 5° of roll angle (**a**) True-Positive (**b**) True-negative (**c**) False-positive (**d**) False-negatives (**e**) RMSE of attitude estimation.

**Figure 9 sensors-20-02388-f009:**
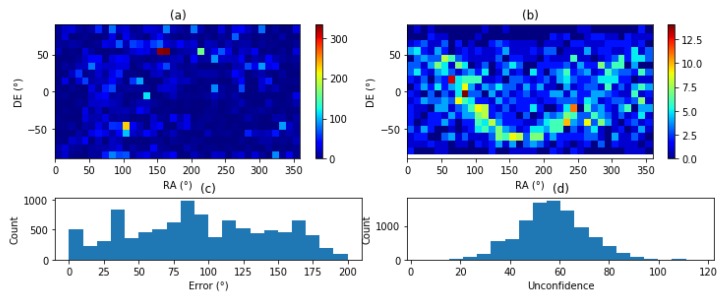
Analysis of wrongly identified frames for every roll angles (**a**) 2D Histogram of feature position, (**b**) Star (mag < 5) 2D histogram, (**c**) Histogram of error (°), (**d**) Histogram of confidence.

**Figure 10 sensors-20-02388-f010:**
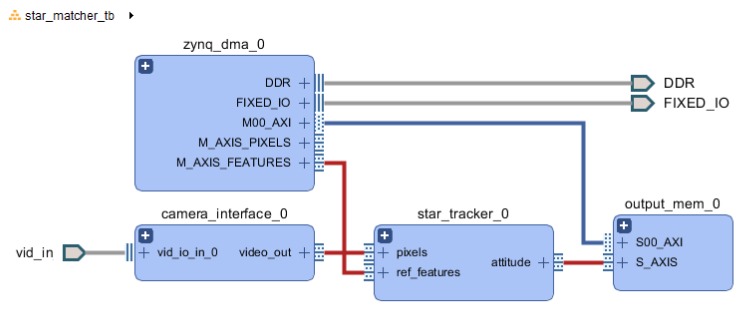
Integration with a Camera on a Zynq system.

**Figure 11 sensors-20-02388-f011:**
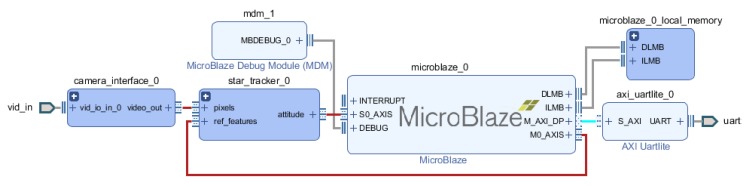
Integration with a camera without a MicroBlaze soft processor.

**Figure 12 sensors-20-02388-f012:**
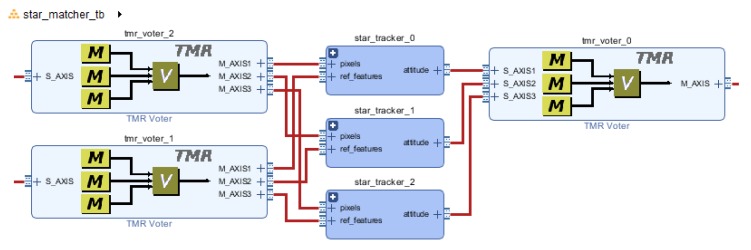
TMR implementaton on Vivado IP Integrator.

**Table 1 sensors-20-02388-t001:** Optical parameters.

*T*	=	0.020 s	Exposure time (50 frames/s)
Qmax	=	14,000 $e^{-}$	Sensor’s Full Well capacity
$Q_{e}$	=	0.55	Sensor’s Quantum Efficiency
$T_{lens}$	=	0.7	Average Lens Transmission
$A_{lens}$	=	17/0.9mm or 25/0.9mm	Lens Aperture and f-number
FOV	=	51.6° or 36.3°	Associated Field of View with given sensor
BW	=	500nm	Lens and sensor bandwidth: 300–900 nm
$e^{-}/{\mathrm{LSB}}_{14}$	=	0.55	Sensitivity, in Electrons per LSB at 14 bits output

**Table 2 sensors-20-02388-t002:** Average center pixel value induced by three stars of apparent magnitude 0, 3 and 5 for 20 ms exposure time with three different PSF.

Optics (Focal / f#, FOV)	Star (Hip. id)	V	Center Pixel Value		
			σ=0.5	σ=1	σ=2
25/0.9 mm, FOV = 36°	α-Lyrae (91262)	+0.03	170,007/$2^{14}$	86,977/$2^{14}$	27,365/$2^{14}$
	γ-Lyrae (93194)	+3.25	8758/$2^{14}$	4480/$2^{14}$	1409/$2^{14}$
	λ-Lyrae (93279)	+4.94	1846/$2^{14}$	944/$2^{14}$	297/$2^{14}$
17/0.9 mm, FOV = 51.6°	α-Lyrae (91262)	+0.03	78,611/$2^{14}$	40,218/$2^{14}$	12,653/$2^{14}$
	γ-Lyrae (93194)	+3.25	4049/$2^{14}$	2071/$2^{14}$	651/$2^{14}$
	λ-Lyrae (93279)	+4.94	853/$2^{14}$	436/$2^{14}$	137/$2^{14}$

**Table 3 sensors-20-02388-t003:** Summary of the proposed star identification algorithm’s robustness against noise injection.

Noise Model	All 2500 Randomly Tested Attitude			Confidence Score < 5		
	RMSE	Mean Error	Confidence Score	%	RMSE	Mean Error
Noiseless	0.000°	0.000°	0.054	100.00	0.000°	0.000°
±0.1°	3.699°	0.134°	0.486	99.92	0.039°	0.035°
±0.2°	3.386°	0.161°	0.949	99.77	0.076°	0.068°
±1.0M	0.000°	0.000°	0.054	100.00	0.000°	0.000°
±0.1°, ±1M	0.040°	0.034°	0.482	99.92	0.038°	0.034°

**Table 4 sensors-20-02388-t004:** Star Identification Accuracy: summary of 10,000 tests from cropped image.

	$\sigma^{2}=0.5$		$\sigma^{2}=1.0$		$\sigma^{2}=1.6$	
Wsize	%Confident	RMSE	%Confident	RMSE	%Confident	RMSE
3	84.24%	0.003878°	98.84	0.000964°	99.23	0.000942°
5	81.66%	0.003821°	98.81	0.000954°	99.20	0.000823°
7	83.27%	0.003861°	98.78	0.000954°	99.14	0.000901°

**Table 5 sensors-20-02388-t005:** Estimation of the impact, magnitude and counter-measure to various noise sources.

Noise Source	Synergy	Effect on Data	Impact	Effet on the Proposed Star Tracker
Shot Noise	↗ noise with low *V*	Noisy stars	+	Noisy detection, centroid, magnitude
Sun and Moon	↗ events with FOV	Occlusion	+	Partial obstruction, lower availability
Sun and Moon	↗ events with FOV	Blooming	+	Reduced availability, increased noise
Dark current	↘ with update rate	Lower readout	–	Noisy detection, centroid, magnitude
Thermal noise	↗ with temperature	Global noise	∼	Noisy detection, centroid, magnitude
Lens distortion	↗ with FOV	Wrong position	∼	Can be compensated
Dead pixels	↗ with FOV	False Detection	∼	Can be excluded
Non ideal PSF	↗ with FOV	Distorted PSF	∼	Compensated with large Wsize
Star motion	↗ with time	Centroid noise	∼	Compensated with an recent database
Angular rate	↗ with exposure	Motion blur	±	Limits the maximum angular rate

**Table 6 sensors-20-02388-t006:** Analysis of star identification rate and accuracy when subjected to a high angular rate with $\sigma^{2}=1.6$.

		Wsize=3		Wsize=5		Wsize=7		Wsize=9	
Angular Rate	Spot Width	Coverage	RMSE	Coverage	RMSE	Coverage	RMSE	Coverage	RMSE
0°/s	0 px	99.4%	0.00091°	99.4%	0.00080°	99.4%	0.00080°	99.4%	0.00081°
5°/s	1 px	99.2%	0.00114°	99.2%	0.00090°	99.1%	0.00091°	99.1%	0.00091°
10°/s	2 px	99.3%	0.00224°	99.5%	0.00117°	99.4%	0.00116°	99.4%	0.00116°
15°/s	3 px	96.7%	0.00504°	99.1%	0.00151°	99.2%	0.00151°	99,2%	0.00153°
20°/s	4 px	87.3%	0.01008°	93.7%	0.00231°	99.3%	0.00194°	99.2%	0.00197°
30°/s	6 px	18.5%	0.03189°	18.6%	0.01163°	28.3%	0.00531°	94.2%	0.00269°

**Table 7 sensors-20-02388-t007:** Analysis of Baffle length for various optical systems with an Earth Aspect Angle of 30°.

Sensor Format	FOV (°)	f/f# (mm)	*a* (°)	α (mm)	$\phi_{T}$ (°)	*L* (mm)
1″	50	17/0.95	25°	8.95	30	161.2
1″	36	25/0.95	18°	13.15	30	104.2
1″	50	16/1.40	25°	5.71	30	102.9
1″	36	25/1.40	18°	8.92	30	70.7

**Table 8 sensors-20-02388-t008:** Resource usage.

IP	Slice (%)	LUT as Logic (%)	LUT as Mem (%)	BRAM (%)	DSP (%)
axis_data_fifo	51 (0.4%)	27 (0.1%)	68 (0.4%)	0.0 (0.0%)	0.0 (0.0%)
extract_roi_3	258 (1.9%)	158 (0.3%)	640 (3.7%)	0.0 (0.0%)	0.0 (0.0%)
feature_extractor_st	1001 (7.5%)	2497 (4.7%)	130 (0.7%)	2.0 (1.4%)	3.0 (1.4%)
feature_matcher_strm	802 (6.0%)	1383 (2.6%)	334 (1.9%)	3.0 (2.1%)	0.0 (0.0%)
star_detector_3	187 (1.4%)	276 (0.5%)	12 (0.1%)	0.0 (0.0%)	1.0 (0.5%)
Total	2281(17.2%)	4341 (8.2%)	1184 (6.8%)	5.0 (3.6%)	4.0 (1.8%)
Available (XC7Z020)	13,300 (100%)	53,200 (100%)	17,400 (100%)	140 (100%)	220 (100%)

**Table 9 sensors-20-02388-t009:** Statistics of database performances for different number of features per frames over 10,000 tests.

*k*	Total	Unique	Size (kB)	Matching Delay (μ s)	Total Time (ms)	Confident	RMSE
1	64,800	3221	83	167	15.0	9004/10,000	0.000299°
2	129,600	6209	161	322	16.6	9586/10,000	0.000301°
3	194,400	9034	235	469	18.1	9738/10,000	0.000309°
5	324,000	14,396	374	748	21.0	9959/10,000	0.000301°
9	583,200	22,505	585	1170	25.3	9990/10,000	0.000295°
15	972,000	32,311	840	1680	30.6	9995/10,000	0.000305°
21	1,360,800	39,158	1018	2076	34.1	9998/10,000	0.000307°

**Table 10 sensors-20-02388-t010:** Comparison between commercially available CubeSat star trackers and the proposed star tracker.

Parameter	MAI-SS Space Sextant	ST400	Proposed
Interface	I2C	RS422/UART/I2C/SPI	RS485/UART/LVDS
FOV	NA	15° × 18°	51° or 36°
Update rate	4 Hz	5 Hz	25–50 Hz
Accuracy	4 arcsec/27 arcsec	10 arcsec	4 arcsec/26 arcsec
Compatible OBC	Any	Any	Any or NanoMind Z7000
Power consumption	1.5 W	0.7 W	0.13W FPGA + 0.4 mW Senor
Weight	282 g	280 g	250 g
Volume	55 cm × 65 cm × 70 cm	48 cm × 57 cm × 89 cm	50 cm × 50 cm × 70 cm
Baffle size	5″	included	70–160 mm
Manufacturer	MAI	Berlin Space Technologies	NA
Retail price	32,500 $	NA	NA

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
