# Peer review of "Design and Simulation of a High-Speed Star Tracker for Direct Optical Feedback Control in ADCS"

_sensors, 2020, doi:10.3390/s20082388_

Round 1

Reviewer 1 Report

Brief Summary

In this manuscript, the authors describe the design and validation of a high-speed star tracker concept. The authors briefly describe the star tracker hardware requirements, a modified algorithm for faster implementation for attitude determination and an FPGA implementation design for the same. 

The paper states the importance of high-speed star trackers and tried to present an interesting solution to this. But the manuscripts lack some clarity and need some major revisions.

Broad Comments

Most of the work on the algorithm is mentioned in the author's previous work - 10.1109/MetroAeroSpace.2018.8453507 . This manuscript is a continuation of that work with a focus on an FPGA implementation strategy of the algorithm and verification. I am not very convinced by the novelty of the feature-description based algorithm, described in the paper. The authors also state that it's based on the existing Pyramid algorithm. A better explanation of the novelty of the algorithm is required. Also, authors could add a basic flowchart outline major process involved in the algorithm.

It's not very clear to me the comparison analysis of high-speed star tracker with the low-speed star tracker with gyro. Please bring in better clarity in the analysis.

The description of FPGA implementation with the vivado block diagram is more applicable to a technical report. I would recommend the authors summarise it further.

Also, I would like the authors to address the following concerns in the manuscript.

- Consider all different noise sources of the detector in the sensitivity and requirement analysis. 
- What are the considerations/constraints for lost in space mode operation for the proposed high-speed tracker?
- Since the proposed star tracker is very wide-field what are the possible implications when Sun/Moon in the FOV. Comment on the robustness in this scenario and please add this also in the manuscript 
- Comment on the robustness during a high rate motion during non- routine mission phases of the satellite (eg: detumbling)

Specific comments

Line 8: Change to- 'terms' of

Line 20: 'Attitude Determination has greatly profited from the use of imaging
sensors in star trackers' - Every star tracker has an imaging sensor. So, I think no need of 'imaging sensors in' in this sentence.

Line 47: provide expansion for IP

Equation 6: Provide explanation/reference on how you reach to this equation. Also explain different terms in this equation such as M, Mα, Lyrae

Line 85-86: Rewrite the sentence, it's not clear. What is the context of 'Teledyne e2v Onyx 1.3 sensor and two different optics: ' here.

Table 1: Provide PSF/FWHM information. Variance - 0 doesn't make much sense in a realistic optical system. And I think it's better to replace the variance with PSF/FWHM throughout the manuscript. Provide focal length/plate scale also in Table 1.

Table 2: What is the effect of different noise sources, such as readout noise, dark noise, etc ? Provide a sensitivity analysis and signal to noise ratio estimation.

Line 100: Provide the relevance of the algorithm explicitly in the first instance.

Line 106: Could authors justify why they have chosen Pyramid algorithm-based approach, provided there could be faster star identification algorithms? A comparison with other potential algorithms would provide a better picture.

line 141: Rephrase- The 'focus' of ....
line 147: Please provide a description of the Wsize x Wsize pixels patch and provide an appropriate reference. And provide a justification why specifically this method?

Equation 10: What the symbol "%" stand for? possibly division? please modify.

Section 3.4:

"This strategy has proven to be extremely effective,
but increases the computation time of the feature matching algorithm by a factor ( N N-k)" -

Can you provide a reference?

I am confused by the representation of the factor (N N-k). Is it (N/(N-k)). please modify. Also, please clearly states what N stands for.

Line 152: Based on what assumptions one can set the threshold value here? Please explain.

Figure 3: I think there is no need for the text 'unique features of 2 to 5 stars' in figure 3.

Table 3: Provide abbreviation for RMSE, either in table or text.

line 180: What is ' L-1 norm distance". 

Equations 8,9,10: provide an explanation for different terms 

Section 3.5.2: What method specifically used to determine the attitude quaternion?

Equation 12: Please rearrange the equation- The representation terms (Is) near the square root symbol is confusing. Do you have a reference or derivation for this equation?

line 201: change to "CPU"

line 208: remove one "a"

Figure 5: Please denote each figure's identifier (a, b, c..) in bold. Its difficult to identify these letters from the figure as of now. The caption of Figure 5 is not very clear, please modify for better clarity of the message. What are the different colours for? please provide the details.

Figure 8: Please provide a more descriptive caption

line 220: Provide abbreviation for BOM
line 235: Provide abbreviation for AXI4 and reference.
line 236: explain 'tlast' or give reference
line 255: remove 'sort'
line 321: Correct typo
line 347: provide abbreviation for BMR, I have noticed many abbreviations are missing throughout the manuscript, please add wherever required.

line 365: Change to authors

Section 5:

I dont think the first two points are very relevant or unique and may be omitted.

The authors could provide details of the proposed high-speed star tracker, such as weight, volume, power and other major parameters (such as pointing accuracy, update rate etc), which are relevant for small satellite platforms. Maybe a table would be better. This would be interesting for the readers.

Reviewer 2 Report

This paper studies the feasibility of high-speed star trackers with modern optics, sensor and computing system. The introduction states the purpose of the paper, and the relation between the paper and the previous works is explained. An algorithm is designed to operate at high speed. The performance of the algorithm is evaluated for the implementation on FPGA.

The following are my specific comments:

(1)It is stated in the introduction that “there are required to use a relatively long exposure time in order to increase the SNR of the image”. For clarity, it is better to give some suggestions for the high-speed star tracker with a short exposure time to increase the SNR.

(2)How is the attitude measurement accuracy of the star tracker guaranteed for the satellite may require wide angle maneuvers with high angular rate? Try to give some explanations.

(3)According to Table 1, the FOV of the designed high-speed star tracker is 51.6°. In my opinion, this FOV is rather large for a high-accuracy optical sensor. To demonstrate the feasibility of the scheme, the authors are suggested to provide a brief explanation about the relation between the FOV and the NEA of a star tracker.

(4)It is concluded in Section 2 that it is possible to detect stars up to an apparent magnitude of 5 at 50 frames per seconds. To illustrate that the proposed attitude determination system achieves a 0.001° accuracy, more works are required to analyze the effect of the exposure time to the centroiding accuracy for a particular source (for example, a star with an apparent magnitude of 5).

(5)Is the imperfections of the optical system taken into consideration in the analysis of the attitude estimation accuracy in Section 3? It is better to give some remarks about this problem.

(6)The authors are suggested to compare the proposed star matching algorithm with other existing algorithms via simulations.

(7)In sub-section 3.5.3, it is stated that the “MEMS rate gyro have a noise of 1°”. I suppose that “have” may be “has”, and “0.1°” may be 0.1°/s or 0.1°/h. Please check.

(8)The characters shown in Figure 6 are somewhat blurry.

(9)It is stated in the conclusion that 99% of the sky presents 3 stars of more with an apparent magnitude lower than 5.0. I suppose that “of” may be “or”. In addition, I wonder how to deal with the attitude estimation problem in the case that there are less than 3 stars in the FOV.

Round 2

Reviewer 1 Report

Thank you, authors, for trying to address my comments. However, I am not satisfied with certain issues and I feel those should be addressed.

My comments on the revised manuscript are as follows:

1) Even though the authors tried to mention the possible implications or effects of different noise sources, I would like to have these included in the actual simulation. The authors state that when the noise levels are above the threshold, the algorithm returns erroneous values and prevent false detection. I would like to see a probability of those false detections in the real simulation where you have incorporated the different noise sources and scenarios. For example, as mentioned in the manuscript, with a 50-degree FOV star tracker, the effect of Sun/Moon in the star tracker FOV, could be very prominent and can significantly affect the efficiency of the proposed instrument. I would suggest authors come up with more realistic simulations (including the possible observational scenarios) and include that as a detailed discussion under section 3.5.3.
As of now only synergies and effects are listed.

2) I still fell a flowchart would be better than figure 6, which is less descriptive anyway. What is "is last?" block in the proposed flow chart, be, please more self-descriptive and lucid. Add a modified and complete flowchart.

3) For me previous comment- "What are the considerations/constraints for lost in space mode operation for the proposed highspeed tracker?"

the following response from authors is not very satisfactory-

"Most star-trackers using a very narrow field of view and time-limitations do rely on a tracking approach. In our case, a lost-in-space search is not a constrain, but a mere consequence of other design choices leading to this algorithm. We could question if a tracking the algorithm could be implemented to increase the update rate and stability, and the authors are fairly certain that at a system level, some strategies and filtering method should be implemented to smooth the estimation of the angular rate, as well as provide an estimation of the attitude whenever the algorithm is inconclusive. A large untold of this manuscript is the context of the integration of this star tracker into an ADCS: the video stream will be split in 4 and the star tracker will be one of 4 subsystems analyzing the video stream as described in the introduction. There will be a visual gyroscope, analyzing the motion from frame to frame, a sun sensor, and a horizon sensor. Those different measurements will then be processed by the ADCS algorithm to control the attitude of the satellite regardless of the dominant input. This will improve our current CubeSat ADCS which performs a sensor fusion between a star tracker, a mems gyro, a horizon sensor, and a sun sensor, by providing a unified sensing platform requiring 2 of the same imaging sensors connected to a single Zynq-based Onboard
Computer without any additional electronics."

- I am not very convinced by the response of authors in this matter. The lost in space mode operation and tracking estimation is critical for the proposed "high-speed start tracker". I would like the authors to consider a detailed discussion on that part in order to increase the credibility of the system. And one of the advantages of this system is possible to reduce the use of other sensors such as gyros...I am surprised to see the author's response that the final ADCS will again have the proposed star trackers along with other sensors and this seems to be a bit contradictory to the aim of this manuscript? Please provide justification.

4)Also regarding the scattering effect due to Sun/Moon could be critical. I think authors should give at least a preliminary design on the baffling system and simulate its effect on the final performance of the system. These things are critical for the proposed high-speed star tracker.

5)I would also suggest authors simulate the cases for different angular rates and come up with the performance of the proposed system. For a high-speed star tracker, the robustness during high rate motion is an interesting feature to be highlighted.

6)All other modifications seem to be appropriate and satisfactory for me.

Reviewer 2 Report

The manuscript has been improved and now warrants publication.

Author Response

Dear Reviewer 2,

Thank you very much.

Best regards,
Marin Mikaël and Bang Hyochoong

Round 3

Reviewer 1 Report

Thank you, authors, for addressing my comments.

Few final suggestions-

1) Since the manuscript doesn't actually describe the hardware implementation of the proposed star tracker- I feel the title should be modified; as of now, it could be a bit misleading. The manuscript mostly covers the design and simulation of the proposed high-speed star tracker. So please modify accordingly (Something like- Design and simulation of a high-speed Star Tracker concept for direct optical feedback control in ADCS).

2)I would suggest authors make the simulation codes available public by putting them in an online repository. And please add the corresponding repository web link in the manuscript. This could be beneficial to the wider community.
